# New Considerations on the Determination of the Apparent Shear Viscosity of Polymer Melt with Micro Capillary Dies

**DOI:** 10.3390/polym13244451

**Published:** 2021-12-18

**Authors:** Wangqing Wu, Ke Zeng, Baishun Zhao, Fengnan Duan, Fengze Jiang

**Affiliations:** 1State Key Laboratory of High-Performance Complex Manufacturing, School of Mechanical and Electrical Engineering, Central South University, Changsha 410083, China; csu_zk95@csu.edu.cn (K.Z.); qustzbs@163.com (B.Z.); dfn0801@163.com (F.D.); 2Institute of Polymer Technology, Friedrich Alexander University Erlangen Nurnberg, Weichselgarten 9, D-91058 Erlangen, Germany

**Keywords:** apparent shear viscosity, micro injection molding, polymer melt, capillary flow

## Abstract

Capillary rheometers have been widely used for the rheological measurement of polymer melts. However, when micro capillary dies are used, the results are usually neither accurate nor consistent, even under the same measurement conditions. In this work, theoretical modeling and experimental studies were conducted for a more profound understanding of the mechanism by which the initial and boundary conditions influence the inaccuracy in the apparent shear viscosity determination with micro capillary dies (diameters: 500 μm, 200 μm, 100 μm). The results indicate that the amount of polymer initially in the barrel, the pre-compaction pressure and the capillary die diameter have a significant influence on the development of the micro scale inlet pressure, which directly determines the accuracy of the measurement at low and medium shear rates. The varying melt compressibility was confirmed to be the main factor directly related to the inaccuracy in the micro scale apparent shear viscosity determination. It is suggested that measures such as reducing the amount of polymer initially in the barrel and increasing the pre-compaction pressure could be used to reduce the measurement inaccuracy.

## 1. Introduction

Microinjection molding (*μ*IM) is considered to be one of the most flexible, reliable and efficient processes to produce thermoplastic polymer microparts. The main fields of application of polymer microparts are seen in the areas of biotechnology [1], as components of optical systems [2], micro gears in micro fluidics, medical technology, electronics or as a micro-electromechanical system [3]. With the rapid development and wide application of micro-injection molding, it is an inevitable trend to apply numerical simulation methods to study the problem of mold filling flow in micro-injection molding [4]. The accuracy of the simulation results depends to a certain extent on the description of the material viscosity at the micro-scale. However, as the dimensions of the parts, and thus the mold cavities, become smaller, some factors normally neglected in conventional injection molding (CIM) may play important roles in determining the flow behavior [5,6,7]. It can be pointed out that existing packages are no longer sufficient to capture all the information needed to describe the *μ*IM process. The inadequacy is attributed to the unavailability of rheological data from microscopic experiments, and the lack of consideration of wall slip, surface tension and other relevant effects in *μ*IM [8,9]. Therefore, the study of the rheological behavior of the polymer in micro scale is one of the aspects related to the technology of micro injection molding that still needs to be studied in depth.

High-pressure capillary rheometers that are based on Poiseuille flow have been the most commonly used for characterizing the rheological properties of polymer melts [10]. As in one typical application, they have been widely implemented for the constitutive characterization and modeling of polymer melts for processing simulations. In general, the operation of a capillary rheometer is as follows: the barrel and die are uniformly heated up to experimental temperature and the polymer pellets are poured into the measuring cavity above the die consecutively using small amounts and manually compressing them afterwards, until the measuring cavity reaches the required amount. The piston is then moved onto the polymer until a pressure of 1 MPa is reached and a waiting time of 600 s follows next to ensure the polymer is completely and uniformly molten. After the melting time is over, the piston moves downward at a constant speed, and the shear viscosity at different shear rates is continuously measured. Using this conventional measurement method, we found that the viscosity of cyclic olefin copolymer (COC) witnessed an increase in a certain shear rate range in 200 μm and 100 μm dies [11]. However, from previous works, viscosity increase is impossible. Chien [12,13] developed a series of micro channels in a die that was embedded in a mold in order to study the rheological behavior of polymer melts under isothermal conditions using an injection molding machine. They found the viscosity of the melt flowing through the micro-channels will be lower than the viscosity in a conventional size channel, and it follows the law of shear thinning. Dealy et al., [14] used a 250 μm die to study the slip and fracture of HDPE, and no increase in viscosity was found. In addition, so far, apart from us, there are no report in the literature about the use of a 100 μm die on capillary rheometers for rheological studies. Further investigations on micro-scale rheological properties are clearly warranted. Therefore, polypropylene (PP) was used by us for viscosity studies at the micro-scale. According to the conventional measurement method, the phenomenon of viscosity increase had not been found when the PP melt is flowing through the 200 μm capillary die. However, in one accidental experiment, the precompression pressure was not set to 1 MPa, but only 0.5 MPa, and the amount of polymer melt in the barrel was less than usual. As a result, the viscosity increased at the medium shear rates (data are not shown here). We speculated that the increase in viscosity at the micro-scale may be caused by the changes in measurement conditions. Affected by the size effect, the measurement conditions of rheometers that have been standardized at the macro-scale may no longer be applicable to the micro-scale. Therefore, with the further exploration of micro-scale rheological behavior, the measurement conditions related to the micro capillary die need to be updated.

The study of the mechanism can deepen understanding of the increase in viscosity with micro capillary die caused by measurement conditions. During the use of a capillary rheometer, due to the isothermal compressibility of the melt, it takes a certain rise time for the melt to flow through the die before reaching steady state flow. Hatzikiriakos and Dealy [15] performed experiments with polyethylenes and showed that the time to reach the steady state increases with the length to diameter (L/D) of the die and the amount of polymer initially in the barrel, and decreases with die diameter and piston speed. However, Leonor [16] pointed out that the contraction ratio and not the die diameter determined the time to reach steady flow. In addition, when a short die having a large diameter is used, the pressure will rise rapidly, and the pressure peak will exceed the pressure during steady state flow [17]. Moreover, a recent study from Du and Sun [18] presented a semi-empirical approach to evaluating rise time of power-law fluids in a capillary rheometer. It can be seen that the rise time is related to factors such as the amount of polymer melt, the diameter of the die, and the piston speed, which can be linked to the measurement conditions with a micro capillary die. These investigations suggest a new idea that the increase in viscosity at the micro-scale may be related to the unsteady flow of the melt. Exploring the process of melt pressure rising from unsteady flow state to steady flow state, it may be possible to obtain the mechanism of increase in viscosity at the micro-scale.

The main purpose of this investigation was to determine the influence of the amount of material in the barrel, the pre-compaction pressure and the die diameter on the accuracy of viscosity data at low and medium shear rates. A mathematical model was established to qualitatively analyze the impact of compressibility on the measurement results. It is shown for the first time that accurate rheological data flowing through micro-scale capillary dies can only be obtained by reducing the amount of material and increasing pre-compaction pressure, which is different from the conventional diameter die measurement method.

## 2. Modeling

The working process of the capillary rheometer was simplified, as shown in Figure 1. The piston moves down at a constant speed, and the polymer melt in the barrel flows out from the die under pressure. When the melt in the barrel reaches a stable flow state, the rheometer collects the pressure data at the entrance of the capillary die and calculates the viscosity at that shear rate.

In order to simplify the rise time process for the polymer melt to reach steady state flow, the assumptions are:the initial pressure in the barrel is 0 Pa;the melt is a Newtonian fluid, its apparent shear viscosity η does not change with the shear rate;the polymer in the barrel is completely melted without bubbles;the pressure in the barrel is the same, and there are no drag losses;

The polymer melt is a slightly compressible fluid [19,20], and the density as a function of pressure can be represented by:(1)$$\rho \left(P\right)=\rho_{0}\left(1+\beta P\right)$$
where ρ is the density at pressure *P*, $\rho_{0}$ is the density at atmospheric pressure, β is the isothermal compressibility coefficient of the polymer melt, and *P* is the gauge pressure.

When the piston moves downwards at a constant speed, part of the melt is extruded from the die and part of the melt is compressed in the barrel. The whole process follows the law of conservation of mass, which can be represented as:(2)$$m_{initial}=m_{barrel}+m_{out}$$

It can be noted that the amount of polymer melt in the die is at least three orders of magnitude lower than that in the barrel, so this part of the amount is ignored.

The principle of the rheometer is based on Poiseuille flow [21]. Therefore, the volume flow rate through the die can be represented as:(3)$$q=\frac{\pi Pr^{4}}{8\eta L_{1}}$$
where *q* is the volume flow rate through the die, *P* is the pressure in the barrel, *r* is the radius of the die, *L*_1_ is the die length and *η* is the viscosity of the melt.

The mass of the melt is equal to the product of volume and density. Substituting Equation (3) into Equation (2), Equation (2) can be re-written as:(4)$$\pi R^{2}L_{0}\rho_{0}=\pi R^{2}L\rho +\rho_{0}\int_{0}^{t}qdt$$
where *L*_0_ is the initial length of polymer in the barrel, *L* is the length of polymer in the barrel at time t, and R is the radius of the barrel. Since the melt is set as a Newtonian fluid in the assumptions, there is no outlet pressure drop. Therefore, the pressure at the die outlet is equal to atmospheric pressure, which means the density of the die exit can be described by $\rho_{0}$.

To simplify the equation parameters, a factor *A* is given as:(5)$$A=\frac{r^{4}}{8\eta L_{1}R^{2}}$$

Combining Equations (1), (3)–(5), Equation (6) is obtained:(6)$$A\int_{0}^{t}Pdt+\left(\beta L-\beta v_{0}t\right)P=v_{0}t$$

Solving the differential Equation (6), the relationship between pressure and time is obtained:(7)$$P\left(t\right)=\frac{v_{0}}{A-\beta v_{0}}\times \left[1-{\left(\frac{L-v_{0}t}{L}\right)}^{\frac{A}{\beta v_{0}}-1}\right]$$

## 3. Experimentation

### 3.1. Material

The polymer used in this work was polypropylene in the form of pellets (PP 5090T, produced by Formosa Plastics Corporation, Taiwan, China). The main properties are shown in Table 1. Due to the perennial humidity in the area, the raw material was dried at 80 °C for 6 h in a dryer before being used in case experimental work was influenced negatively by absorbed moisture.

### 3.2. Equipment

Apparent shear viscosity of PP was determined by a high-pressure capillary rheometer, $G\ddot{o}\mathrm{ttfert}$ RG50, with piston velocity ranging from 0.0001 mm/s to 40 mm/s and maximum operation temperature of 400 °C. The temperature accuracy of the instrument is 0.1 °C, and the data collection position accuracy is $1.6\times 10^{-5}$ mm. Three capillary dies manufactured by $G\ddot{o}\mathrm{ttfert}$ with different diameters (0.1 ± 0.005 mm, 0.2 ± 0.005 mm, 0.5 ± 0.005 mm, material: Tungsten Carbide) and length of 5 mm were used to study the accuracy of viscosity measurement at low and medium shear rates.

### 3.3. Methodology

#### 3.3.1. Apparent Shear Viscosity Measurement

Most polymers are pseudo-plastic fluids (non-Newtonian fluids) in the melt state, thus the Rabinowitsch–Weissenberg correction [22] is used for correcting apparent shear rate through capillary die. Also, due to the entrance effects caused by the viscoelasticity of polymers, the Bagley correction [23] should be applied to obtain the real shear stress. In this study, however, the focus is the pressure–time curve to reach steady state flow at a certain shear rate, which means that the corrections can be ignored. Therefore, the rheological curves being studied in this paper are the apparent viscosity and apparent stress curves. In each die, the PP melt was investigated in the low and medium shear rate range (around 60–10^3^ s^−1^) at a temperature of 220 °C.

There are three experimental parameters, namely the die diameter, the amount of polymer initially in the barrel and the pre-compaction pressure. The amount of polymer initially in the barrel is defined by weight, not the length of the barrel occupied by polymer. Due to differences in melt pressure state with the same length of barrel, the amount might be different. What needs illustration is that the definition of pre-compaction pressure is the pressure in the barrel when the piston starts to move, not the precompression pressure applied by the piston during melting stage. Because the melting time is required after entering the barrel, the precompression pressure will change constantly during this period. Therefore, the pre-compaction pressure is chosen as one of the investigating variables in this work. In addition, the pre-compaction pressure of the PP melt in the barrel cannot be precisely defined among measurements because it is a real-time data point recorded by the rheometer once the piston starts moving. Hence, it is reasonable to ignore the slight deviation of the pre-compaction pressure among measurements. Table 2 summarizes the value range of each parameter.

#### 3.3.2. Compressibility Measurement

The pressure close to the capillary entrance was measured with a pressure transducer as a function of time, with data collected every 0.4 s. Experiments to measure polymer isothermal compressibility were also run in the capillary rheometer by using a flat plug instead of a die. The results of density as a function of pressure are given in Figure 2.

It is clear that the dependence of the density on pressure is not linear in the whole range studied, especially in the range of 0–5 MPa. This result is similar to Leonor’s finding [16]. The power function was used to fit the density versus pressure curve in the range of 0–10 MPa, and the coefficient of determination was as high as 99%.

All the experiments were carried out at a temperature (T) of 220 °C.

## 4. Results and Discussion

### 4.1. Size Effect of the Micro Capillary Dies

#### 4.1.1. Apparent Shear Viscosity of Polymer Melt Flowing through 500 μm Capillary Die

Figure 3 shows the rheological curves of PP melt flowing through the 500 μm capillary die with shear rates ranging from 60 s^−1^ to 700 s^−1^, and the different curves have different amounts of polymer and pre-compaction pressure. For example, “6 g–0.11 MPa” means that the amount of polymer initially in the barrel is 6 g and the record real pre-compaction pressure is 0.11 MPa, when the pre-compaction pressure is defined as 0.1 MPa (Section 3.3.1). The apparent shear viscosity of PP melt under different amounts of material decreases with the increase of shear rate, as indicated in Figure 3a, and follows the shear thinning law. In addition, the viscosity curves for different amounts of material are completely overlapped. Correspondingly, the apparent shear stress of PP melt also increases with the increase of shear rate (Figure 3b). Similarly, when keeping the amount of material in the barrel the same and changing the pre-compaction pressure, as shown in Figure 3c,d, the PP melt follows the shear thinning law, and the rheological curves at different pre-compaction pressures are also completely overlapped.

Figure 4 shows the time to reach the pressure sensor collection point of PP melt from unsteady to steady flow at $\dot{\gamma }=60\;s^{-1}$, and the piston moving at a constant speed throughout the process. The data are derived from Figure 3a,c. As shown in Figure 4a, when the pre-compaction pressure is 0.1 MPa, the pressures for different amounts of material all show a slow rise first, then a rapid rise, and finally reach a stable state. The pressure readings collected by the rheometer are all 1 MPa, which intuitively shows that the viscosity measurement results are not affected by the amount of material in the barrel through a 500 μm capillary die. However, the rise times to reach a steady state are different. The rise times collected by the rheometer are 921 s at 6 g, 1280 s at 12 g and 1640 s at 18 g. This means that the greater the amount of polymer in the barrel, the longer the rise time, and the relation between them is not linear, which is contrary to the results of Dealy [15]. As shown in Figure 4b, when the pre-compaction pressure is about 0.3 MPa, the rise time is significantly reduced, only 361 s, which is 2.5 times shorter than at the pre-compaction pressure of 0.1 MPa. It is noteworthy that the pressure with a pre-compaction pressure of 0.35 MPa rises sharply at the beginning and, as time goes by, the rise slows down, and finally reaches a steady state. Obviously, this is different from the pressure rise curve (s-curve) when the pre-compaction pressure is 0.1 MPa.

Theoretical, when the rheological experiment is carried out under the same die at the same temperature, the viscosity curve is unique. The experimental groups with different pre-compaction pressures and amount of material can be compared and verified with each other as illustrated in Figure 3a,c. Therefore, the above results show that neither the amount of material nor the pre-compaction pressure will affect the accuracy of the rheological measurement of PP melt in the range of low and medium shear rates for a 500 μm capillary die, which means the pressure data are the data under steady state flow of the melt. However, reducing the amount of material in the barrel and increasing the pre-compaction pressure can shorten the rise time.

#### 4.1.2. Apparent Shear Viscosity of Polymer Melt Flowing through 200 μ m Capillary Die

Figure 5 shows the rheological curves of PP melt flowing through the 200 μm capillary die with shear rate ranging from 60 s^−1^ to 700 s^−1^. Both the amount of material and the pre-compaction pressure have an impact on the accuracy of the rheological measurement of the PP melt, which is different from the 500 μm capillary die. As shown in Figure 5a, when the pre-compaction pressure is 0.5 MPa, the apparent shear viscosity does not follow the shear thinning law with the increase of the shear rate. When the amount of material is 4 g and 12 g, the viscosity decreases from 60 s^−1^ to 100 s^−1^. Then, the viscosity increases sharply when the shear rate rises to 200 s^−1^, and decreases from 200 s^−1^ to 700 s^−1^. Furthermore, when the amount of material is 18 g, the viscosity decreases from 60 s^−1^ to 600 s^−1^, but the viscosity again coincides with that for 12 g of material at $\dot{\gamma }=700\;s^{-1}$.

It can be seen from the stress curve that the shear stress of the first few collection points does not increase with the shear rate, especially when the amount of material is 18 g (Figure 5b). This means that the pressure read by the pressure sensor is basically the same. Subsequently, the shear stress increased significantly, which corresponds to an increase in apparent shear viscosity. This result is similar to the micro-scale viscosity properties of the COC melt in Lu’s work [11]. Lu explained that the increase of apparent shear viscosity is because of the entanglement of molecular chains at the micro scale. However, this may be caused by the measurement conditions, otherwise it is impossible to have different viscosity data for the same die. This point of view will now be analyzed.

The rheological measurement result is not only affected by the amount of material, but also by the pre-compaction pressure. When the amount of material is 4 g, different pre-compaction pressures will have different viscosity curves (Figure 5c). When the pre-compaction pressure is 0.1 MPa and 0.5 MPa, the viscosity curve first decreases from 60 s^−1^ to 100 s^−1^ with increase of shear rate, then increases at $\dot{\gamma }=200\;s^{-1}$, and finally decreases from 200 s^−1^ to 500 s^−1^. Actually, the apparent shear viscosity rises because the shear stress remains constant at the beginning and suddenly increases at $\dot{\gamma }=200\;s^{-1}$ (Figure 5d). However, when the pre-compaction pressure is 1.2 MPa and 1.5 MPa, the apparent shear viscosity curve follows the shear thinning law, and the two curves completely overlap. An accurate PP melt rheological curve is obtained under these conditions with the 200 μm capillary die. However, when the amount of material is 18 g, no matter what the pre-compaction pressure is, rheological data under steady state flow cannot be obtained in the initial shear rate range (Figure 5e,f). These results indicate that rheological data under steady state flow with a 200 μm capillary die can only be obtained by reducing the amount of material and increasing the pre-compaction pressure. Figure 6 shows the pressure rise curve of the 200 μm capillary die measured by the rheometer in the shear rate range from 60 s^−1^ to 200 s^−1^. The data are derived from Figure 5a,c, and the marked points on the curves represent the pressure data collected by the rheometer at the corresponding shear rate.

When the pre-compaction pressure is 0.5 MPa and the amount of material is 4 g and 12 g (comparing “4 g–0.58 MPa” and “12 g–0.58 MPa” in Figure 6a), the first three pressure readings ($\dot{\gamma }=60\;s^{-1},\dot{\gamma }=80\;s^{-1},\dot{\gamma }=100\;s^{-1}$) were taken within 5 min, and the pressure data are basically the same. However, it took at least 25 min to measure the fourth pressure point, and the pressure rise curve showed an s-shape, similar to measuring the viscosity through the 500 μm capillary die. Moreover, the more material in the barrel, the longer it takes to reach steady state flow, which is similar to the pressure rise curve of the 500 μm capillary die. Note that, when the amount of material is 18 g, the four pressure points were measured in five minutes, and it can be seen from Figure 5b that, until the shear rate is 600 s^−1^, the pressure is basically around 0.47 MPa.

In a pressure collection cycle, if the take-over tolerance (pressure fluctuations in a pressure collection cycle) is less than 1%, it is considered that the polymer melt has reached steady state flow. If not, the rheometer will enter the next cycle until steady state flow is reached. The apparent shear rate is a function of piston velocity, which can be represented as:(8)$${\dot{\gamma }}_{app}=\frac{4q}{\pi r^{3}}$$
where *r* is the radius of die and *q* is the volume flow rate through the die.

Since the melt is extruded by the piston at a constant speed in the barrel, the flow rate can also be expressed as:(9)$$q=\pi R^{2}v_{0}$$
where *R* is the radius of the barrel and *v*_0_ is the speed of the piston.

Combining Equations (8) and (9), Equation (10) is obtained:(10)$${\dot{\gamma }}_{app}=\frac{4R^{2}v_{0}}{r^{3}}$$

Equation (10) shows that the piston speed is proportional to the cube of the die radius at a given shear rate. Therefore, with the same shear rate, the piston speed for the 200 μm diameter die is only 0.054 times that of the 500 μm diameter die, and the piston speed for the 100 μm diameter die is only 0.008 times that of the 500 μm diameter die. The barrel radius of the rheometer is 7.5 mm in this work, therefore, at a shear rate of 60 s^−1^, the piston speed is 0.00417 mm/s for the 500 μm diameter die, while the piston speed is 0.00027 mm/s for the 200 μm diameter die. Combining Figure 4a and Figure 6a, it can be seen that, when the pre-compaction pressure is small (“12 g–0.11 MPa” in Figure 4a and “12 g–0.58 MPa” in Figure 6a), the pressure rises slowly at first. However, when the melt flows through the 200 μm capillary die, the pressure rises less than 1% in the first cycle due to the low piston speed, causing the rheometer software to misunderstand that the melt pressure has reached steady state flow, and to read the pressure under unsteady state. When the rheometer enters the next shear rate, the pressure under the unsteady flow is still read. Until the shear rate is 200 s^−1^, the pressure rise in a single cycle under the unsteady flow state is greater than 1%, the true steady state flow pressure data can be obtained, as shown in Figure 6a. However, the method of reducing the take-over tolerance cannot solve the problem. As shown in Table 3, the take-over tolerance of the “4 g–0.58 MPa” group is 0.09%. As a result, the take-over tolerance setting value should be less than 0.09%. This will not only increase the measurement time but, even if the polymer melt reaches steady state flow, the pressure fluctuation may not be less than 0.09%.

When the amount of material is 4 g and the pre-compaction pressure is 1.22 MPa and 1.56 MPa (see Figure 6b), the pressure rise in a single cycle under the unsteady flow state is greater than 1%, so the pressure is recorded when the melt actually reaches steady state flow, as shown in Figure 6b. Similarly, the pressure rise curve is analogous to the 500 μm capillary die with the pre-compaction pressure set to 0.3 MPa (see Figure 4b). These results clearly show the influence of the amount of material and pre-compaction pressure on the accuracy of rheological data.

#### 4.1.3. Apparent Shear Viscosity of Polymer Melt Flowing through 100 μm Capillary Die

Limited by the minimum speed of the rheometer piston, the shear rate of PP melt flowing through the 100 μm capillary die starts from 200 s^−1^, and the shear rate ranges from 200 s^−1^ to 800 s^−1^. As indicated in Figure 7, this inaccurate measurement becomes more obvious as the die diameter is reduced to 100 μm. When the amount of material is 18 g, no matter what the pre-compaction pressure is, accurate pressure data cannot be obtained in the range of low and medium shear rates (Figure 7a,b and Figure 8a). This fully illustrates that the pressure take-over tolerance in one cycle is less than 1% during the experiment, and then the wrong pressure value is collected. Moreover, when the amount of material is 4 g and the pre-compaction pressure is greater than 1.5 MPa, the viscosity drops in a zigzag manner, while the stress curve rises in a stepped shape with the shear rate. This is because, when the shear rate rises by 100 s^−1^, the piston speed only rises by $5.56\times 10^{-5}\;\mathrm{mm}/s$. The piston speed interval between the two shear rates is too small, resulting in a take-over tolerance of less than 1% when measuring the pressure at the next shear rate, and problematic pressure data are collected. Only if the piston speed is further increased can the take-over tolerance in one cycle be greater than 1% (Figure 8b). This means that, when the die diameter is reduced to 100 μm, it is not appropriate to set the pressure take-over tolerance to 1%, even if the amount of material is reduced and the pre-compaction is increased.

### 4.2. Theoretical Interpretations

The above analysis shows that the increase in viscosity is caused by the wrong measurement conditions, and the inaccurate rheological data will be obtained. The mechanism of this phenomenon is studied by the mathematical model.

There are four variables, namely piston speed, v_0_, die coefficient, A, the isothermal compressibility coefficient, β, and the length of the barrel occupied by polymer, L. It is assumed that the melt is a Newtonian fluid, which means that its viscosity does not change with shear rate. The constants involved in the theoretical analysis are listed in Table 4, and different experimental parameters involved in the theoretical model are listed in Table 5. Note that the length of the barrel occupied by polymer and the amount of polymer in the barrel represent the same parameter, and the length is convenient for calculation. Additionally, the apparent shear viscosity value in this work is from the apparent viscosity of PP through the 500 μm capillary die at $\dot{\gamma }=60\;s^{-1}$, and the isothermal compressibility coefficient, β, is approximately $10^{-9}\;{\mathrm{Pa}}^{-1}$ [24].

Combining Equation (7), Table 3 and Table 4, the theoretical rise time to reach steady flow state for different experimental parameters is shown in Figure 9. Obviously, when the isothermal compressibility coefficient, β, is constant, the slope of the pressure rise curve at time 0 is the largest under different parameter combinations, which means that the pressure rises fastest at the beginning. Therefore, it is impossible to have inaccurate measurement data in this case, even if the die diameter is reduced to 200 μm or 100 μm.

A flat plug was used instead of a die to measure polymer isothermal compressibility by the capillary rheometer. As shown in Figure 2, the isothermal compressibility coefficient of the polymer melt is expressed by the slope. It is clear that the isothermal compressibility is variable in the range of 0–10 MPa, but basically remains unchanged in the range of 10–100 MPa, which is similar to Leonor’s study [16]. This might be caused by the presence of bubbles in the polymer in the barrel. The polymer pellets are poured into the barrel above the flat plug by the conventional measurement methods. After the polymer was completely melted, the flat plug was disassembled and the melt in the barrel was extruded, as shown in Figure 10. The melt is divided into two parts as it falls from the barrel, and there are no bubbles in the first stage close to the die (see Figure 10a), while bubbles can clearly be seen in the second stage (see Figure 10b). This implies that the conventional measurement method cannot completely squeeze out the bubbles, and ultimately affects the isothermal compressibility coefficient. Obviously, there are no bubbles entering the die during the measurement, nor will it change the viscosity of the melt flowing through the die. According to Equation (3), when the volume flow rate, the die parameters and the viscosity do not change, the pressure under steady state flow will not change either. Therefore, a small number of bubbles will not affect the accuracy of the measurement results.

In order to better fit the relationship between density and pressure in the range of low and medium shear rates, the density–pressure curve in the range of 0–10 MPa was intercepted, and the power function was used to fit the curve, which can be represented by:(11)$$\rho P=a\times P^{b}+c$$
where *a*, *b*, *c* are the parameters that need to be fitted. The fitting results of the power function are shown in Figure 2b, Table 6, and the coefficient of determinations was as high as 99%. Therefore, the relationship between PP melt density and pressure is represented by:(12)$$\rho P=12\times P^{0.216}+727$$

Combining Equations (3)–(5), (12), the relationship between pressure and rise time is obtained:(13)$$A\int_{0}^{t}Pdt+\left(\frac{12L}{\rho_{0}}-\frac{12}{\rho_{0}}v_{0}t\right)\times P^{0.216}=v_{0}t$$

Equation (13) is a nonlinear equation with no analytical solution. A numerical analysis method with MATLAB was used to obtain a numerical solution, and then to obtain the relationship between pressure and time. Combining the parameters in Table 4 and Table 5, the new theoretical pressure rise curves under different parameters are shown in Figure 11. When the pressure starts to rise from 0 MPa, the curves under different parameter combinations are all s-shaped, which is completely consistent with the experimental phenomenon. It means that the pressure rise curves are affected by the varying isothermal compressibility. Note that the time and steady state flow pressure of the model results are different from the actual ones due to the assumptions made in the model, for example, that the melt is Newtonian fluid, the pressure in the barrel is the same and the density of the melt flowing through the die is equal to the atmospheric pressure. However, this does not affect the qualitative analysis.

As the diameter of the die becomes smaller, it takes longer for the polymer melt to reach steady state flow, and the longer the low pressure stage becomes at the same shear rate, as shown in Figure 11a. This caused the rheometer software to incorrectly read the pressure under unsteady flow for melt flow through the 200 μm capillary die or 100 μm capillary die. Therefore, increasing the pre-compaction pressure means that the melt pressure in the barrel passed the low pressure stage, which not only shortens the time to reach the steady state, but also avoids the pressure rise in a single cycle under the unsteady flow state being less than 1%. As the amount of material in the barrel increases, the polymer melt spends longer time in the low pressure stage (Figure 11b). Therefore, when the amount of material is 18 g, no matter what the pre-compaction pressure is, accurate rheological data flowing through 200 μm capillary die cannot be obtained in the initial shear rate range (Figure 5e,f). In addition, the increase of shear rate can shorten the time in the low pressure state (Figure 11c), and accurate rheological data can be obtained at a higher shear rate (Figure 6a,b).

The above results clearly show the isothermal compressibility of the melt in the barrel is not constant, and ultimately affects the pressure rise curve. Inaccurate rheological data will not be caused when the isothermal compressibility is constant, because the pressure rises fastest in the initial stage. In short, accurate rheological data flowing through micro-scale capillary dies can only be obtained by reducing the amount of material and increasing pre-compaction pressure.

Finally, viscosity curves of the PP melt flowing through 200 μm capillary and 100 μm capillary dies under different take-over tolerances were measured, as shown in Figure 12. It can be observed that the viscosity curves basically coincide under different take-over tolerance settings when polymer melt flows through a 200 μm capillary die. However, higher viscosity values of melt flowing through a 100 μm capillary die are presented when the take-over tolerance is set to 0.5%. This is because the piston speed is very small using a 100 μm capillary die, which means the melt has not reached steady state flow at take-over tolerance of 1%. Based on the above results, the amount of material was set as 3.5 g, the pre-compaction pressure as 4.0 MPa and the take-over tolerance as 0.5%, and finally the rheological curve for the 100 μm capillary die was obtained at low and medium shear rates (Figure 13).

## 5. Conclusions

The effects of the amount of polymer initially in the barrel and the pre-compaction pressure on the rheological measurement with micro capillary dies were investigated. The results show that accurate rheological data can only be obtained by reducing the amount of material and increasing pre-compaction pressure. When measuring the viscosity of melt flowing through a 100 μm capillary die, it is also necessary to reduce the take-over tolerance setting. In addition, the establishment of a mathematical model provides a qualitative analysis of the causes of inaccurate measurement. Because there are a small number of bubbles in the barrel, the isothermal compressibility of the melt is not constant, and ultimately affects the pressure rise curve. In consequence, during the slowly rising pressure stage, the rheometer software misunderstands that the melt pressure has reached steady state flow, and reads the pressure under unsteady flow. Finally, although the bubbles have an effect on the viscosity measurement with different diameter dies, inaccurate measurement only appears in the 200 μm and 100 μm dies. This means that the viscosity measurement method with a micro-scale die is different from that of conventional dies.

## Figures and Tables

**Figure 1 polymers-13-04451-f001:**
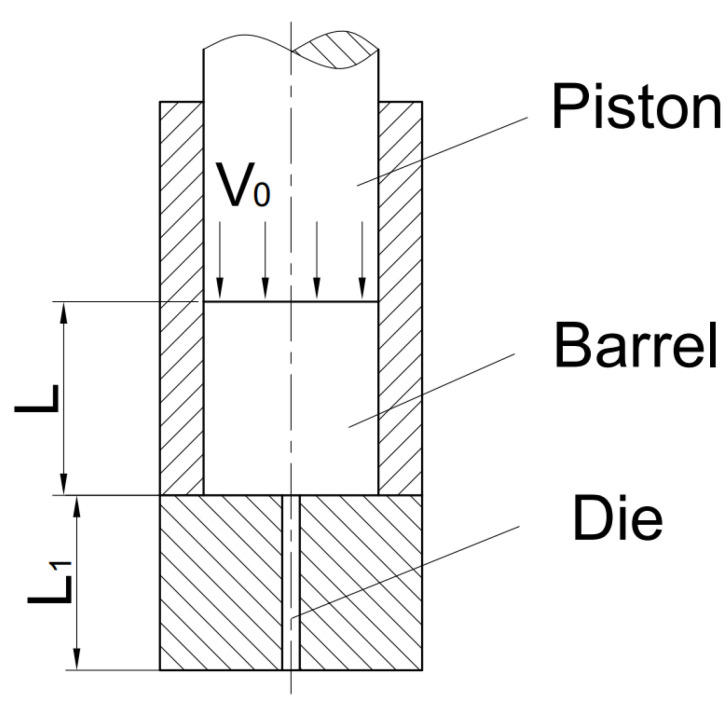
Schematic diagram of the working process of the high-pressure capillary rheometer.

**Figure 2 polymers-13-04451-f002:**
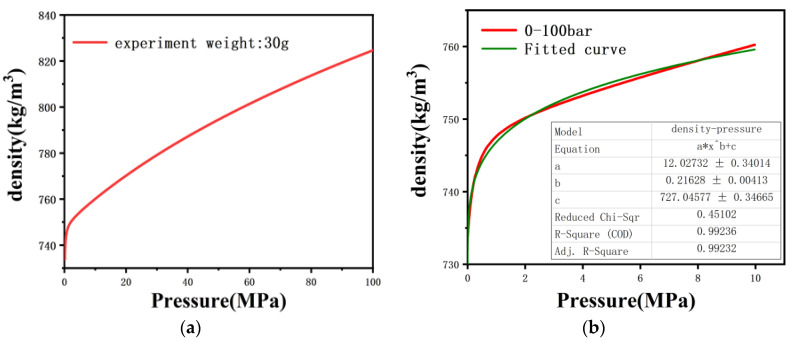
Density as a function of pressure for PP: (**a**) 0–100 MPa and (**b**) 0–10 MPa with fitted curves.

**Figure 3 polymers-13-04451-f003:**
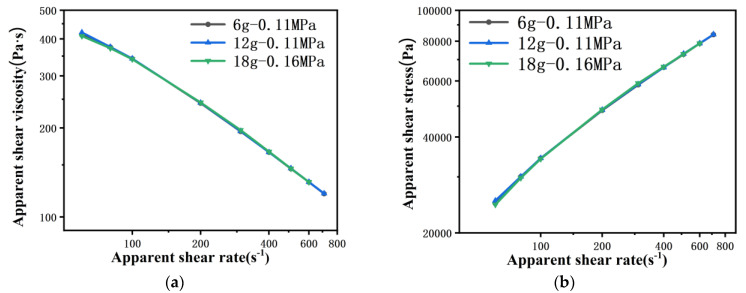
Rheological curves of polypropylene (PP) melt flowing through a 500 μm capillary die with different amounts of material (**a**) viscosity curve and (**b**) stress curve, with different pre-compaction pressures (**c**) viscosity curve and (**d**) stress curve.

**Figure 4 polymers-13-04451-f004:**
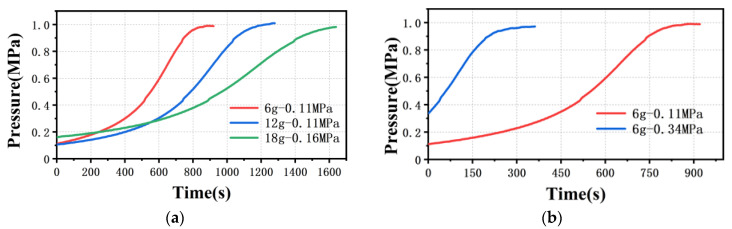
Time to reach the pressure sensor collection point (**a**) with different amounts of material and (**b**) with different pre-compaction pressures ($d_{die}=500\;\mu m$, $\dot{\gamma }=60\;s^{-1}$).

**Figure 5 polymers-13-04451-f005:**
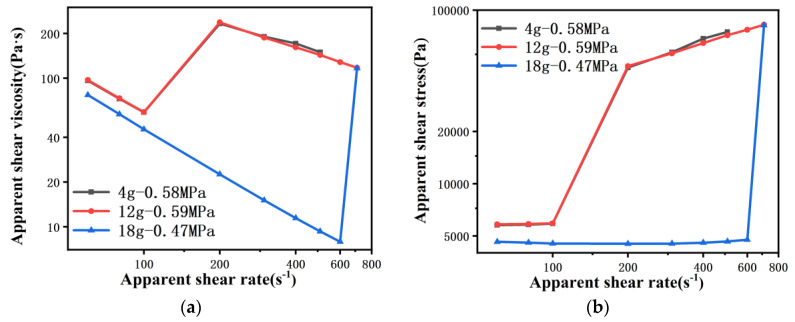
Rheological curves of PP melt flowing through a 200 μm capillary die: with different amounts of material (**a**) viscosity curve and (**b**) stress curve, different pre-compaction pressures at 4 g (**c**) viscosity curve and (**d**) stress curve, different pre-compaction pressures at 18 g (**e**) viscosity curve and (**f**) stress curve.

**Figure 6 polymers-13-04451-f006:**
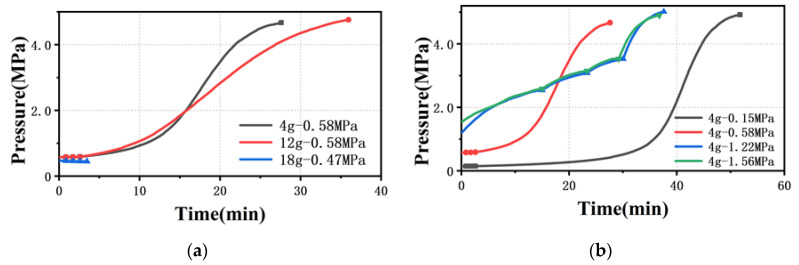
Time to reach the pressure sensor collection point (**a**) with different amounts of material and (**b**) with different pre-compaction pressures. ($d_{die}=200\;\mu m,\;\dot{\gamma }=60\;s^{-1},\dot{\gamma }=80\;s^{-1},\dot{\gamma }=100\;s^{-1},\dot{\gamma }=200\;s^{-1})$.

**Figure 7 polymers-13-04451-f007:**
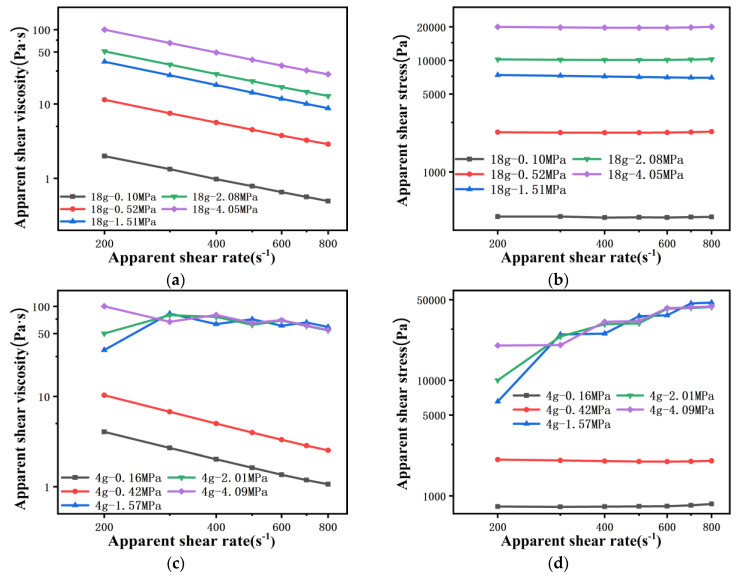
Rheological curves of PP melt flowing through a 100 μm capillary die: the amount of material is 18 g (**a**) viscosity curve and (**b**) stress curve, the amount of material is 4 g (**c**) viscosity curve and (**d**) stress curve.

**Figure 8 polymers-13-04451-f008:**
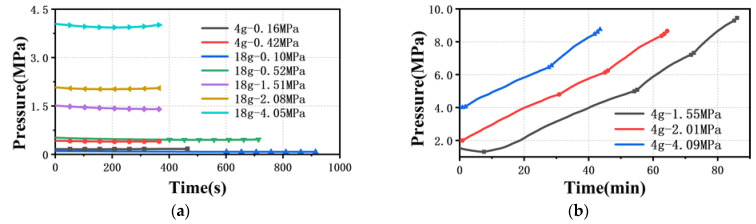
Time to reach the pressure sensor collection point (**a**) with different amounts of material and (**b**) with different pre-compaction pressures. ($d_{die}=100\;\mu m)$.

**Figure 9 polymers-13-04451-f009:**
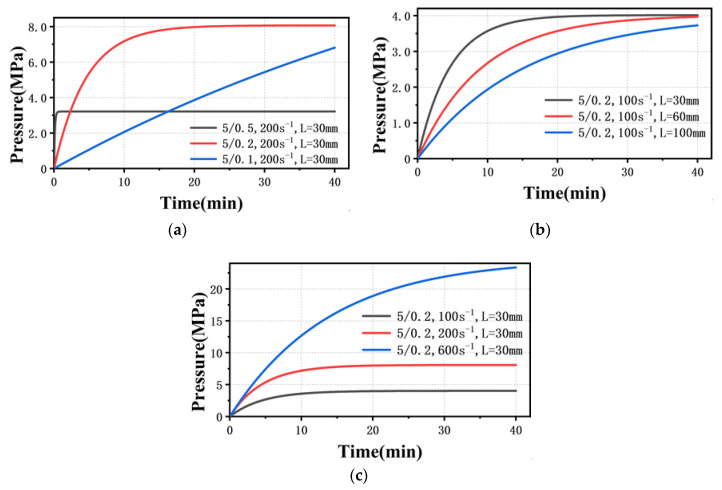
Theoretical pressure rise curve with different parameters (**a**) different amounts of material (**b**) different die sizes (**c**) different shear rates.

**Figure 10 polymers-13-04451-f010:**
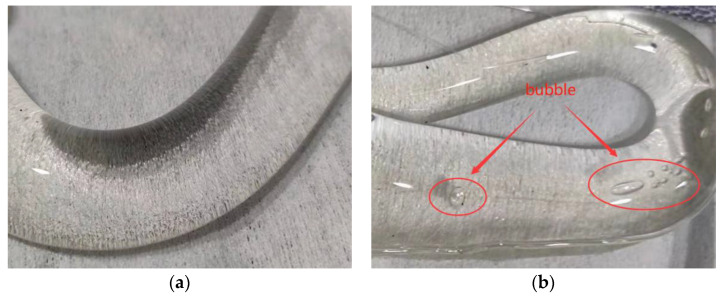
Melt extruded from the barrel (**a**) first stage (**b**) second stage.

**Figure 11 polymers-13-04451-f011:**
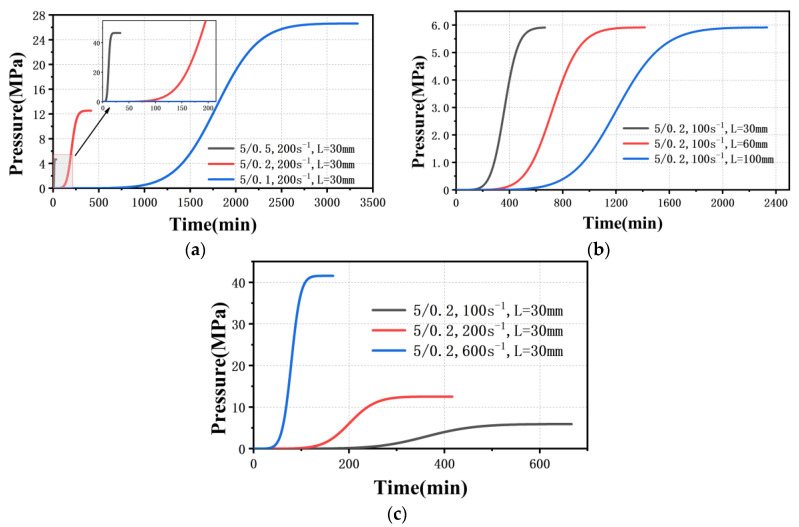
Theoretical pressure rise curves under different parameters with changeable isothermal compressibility (**a**) different amounts of material (**b**) different die sizes (**c**) different shear rates.

**Figure 12 polymers-13-04451-f012:**
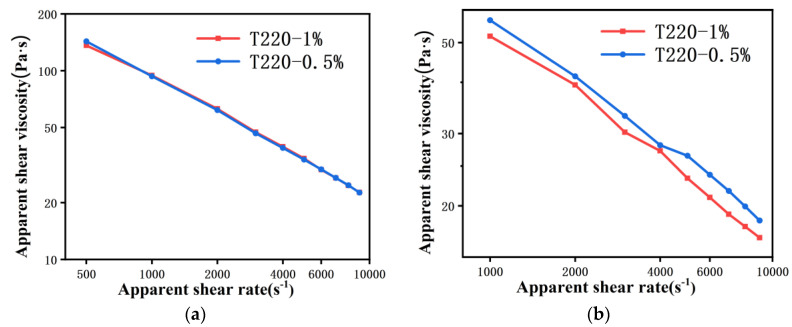
Apparent viscosity curves with different take-over tolerances (**a**) 200 μm capillary die (**b**) 100 μm capillary die.

**Figure 13 polymers-13-04451-f013:**
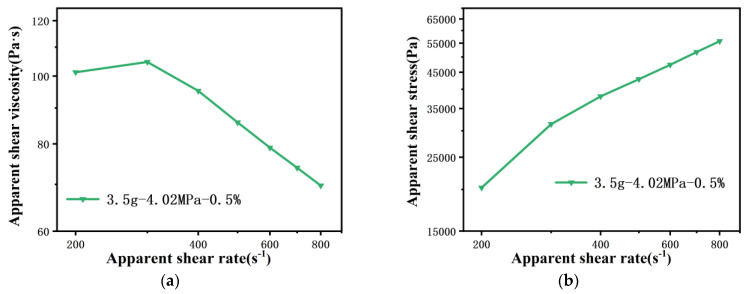
Rheological curves of PP melt flowing through a 100 μm capillary die under 4 g−4.02 MPa−0.5% (**a**) viscosity curve and (**b**) stress curve.

**Table 1 polymers-13-04451-t001:** Polypropylene (PP) material properties.

Properties	Unit	Test Method	Typical Value
Melt Index	g/10 min	ISO1133	15
Density	g/cm^3^	ISO1183	0.90
Melting Point	°C	DSC	146
Heat Deflection Temperature	°C	ISO75	95
Vicat Softening Temperature	°C	ISO306	125
Tensile Strength at Yield	MPa	ISO527	28
Tensile Elongation at Break	%	ISO527	300
Rockwell Hardness	R scale	ISO2039	98
Flexural Modulus	MPa	ISO178	1050
Mold Shrinkage	%	FPC Method	1.3–1.7

**Table 2 polymers-13-04451-t002:** The value range of each experimental parameter.

Parameter	Value Range
die diameter (μm)	500, 200, 100
the amount of polymer initially in the barrel (g)	4, 6, 12, 18
pre-compaction pressure (MPa)	0.1, 0.3, 0.5, 1.2, 1.5, 2.0, 4.0
experiment temperature (°C)	220
shear rate (s^−1^)	60, 80, 100, 200, 300, 400, 500, 600, 700, 800

**Table 3 polymers-13-04451-t003:** The take-over tolerance of different parameter combinations through a 200 μm capillary die.

Parameter Combinations	60 s^−1^	80 s^−1^	100 s^−1^
4 g−0.15 MPa	0.36%	0.42%	0.24%
4 g−0.58 MPa	0.09%	0.24%	0.77%
12 g−0.58 MPa	0.38%	0.31%	0.53%
18 g−0.47 MPa	0.66%	0.55%	0.59%

**Table 4 polymers-13-04451-t004:** The constants involved in the theoretical analysis.

Die Size (mm)	$A\;(\mathrm{mm}\cdot s^{-1}\cdot$ ${\mathrm{Pa}}^{-1})$	η (Pa·s)	$\beta \;({\mathrm{Pa}}^{-1})$
5/0.5	4.34 × 10^−9^	400	10^−9^
5/0.2	1.11 × 10^−10^		
5/0.1	6.94 × 10^−12^		

**Table 5 polymers-13-04451-t005:** Different experimental parameters involved in the theoretical model.

Category	The Comparison Parameters	The Same Parameters
Different amounts of material	L = 30 mm	5/0.2 (mm) $\dot{\gamma }=1$00 s^−1^
	L = 60 mm	
	L = 100 mm	
Different die sizes	5/0.5 (mm)	L = 30 mm $\dot{\gamma }=2$00 s^−1^
	5/0.2 (mm)	
	5/0.1 (mm)	
Different shear rates	$\dot{\gamma }=1$00 s^−1^	5/0.2 (mm) L = 30 mm
	$\dot{\gamma }$ = 200 s^−1^	
	$\dot{\gamma }\;$= 600 s^−1^	

**Table 6 polymers-13-04451-t006:** The fitting results between PP melt density and pressure.

Parameters	$a\;(\left(kg/m^{3}\right)\cdot MPa^{-1})$	b	$c\;(\mathrm{kg}/m^{3})$
Value	12	0.216	727

## Data Availability

The raw/processed data required to reproduce these findings cannot be shared at this time due to technical or time limitations.

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
