# Peer review of "New Considerations on the Determination of the Apparent Shear Viscosity of Polymer Melt with Micro Capillary Dies"

_polymers, 2021, doi:10.3390/polym13244451_

Round 1

Reviewer 1 Report

The experimental determination of the shear viscosity of polymers is an important task in the microscale, when the capillary rheometers with die widths down to 0,1 mm are used. In the paper, a mathematical model of such measurement process was constructed and its solution was applied to explaining the experimentally observed behaviour for polypropylene. The model calculations involved either constant compressibility (linear dependence of density on the pressure) or pressure-dependent compressibility (non-linear dependence of density on the pressure, fitted to the results of measurements). The rheological measurements performed by the Authors involved various die diameters, amounts of polymer in the barrel, pre-compaction pressure and shear rates.

The conclusions drawn involve the recommendations as to the polymer amount and pre-compaction pressure used in rheological measurement with micro capillary dies to minimize the errors.

The paper reports interesting results, useful for the community. I recommend it for publication in Polymers journal, just after the Authors consider the minor points listed below:

Page 2, line 58-59: the sentence “However, from previous works, viscosity increased is impossible.” would be corrected (increased->increase).

Page 3, equation 1: I wonder whether the statement that P in this equation is absolute pressure is correct. Because ρ0 is the density under atmospheric pressure and the equation gives ρ=ρ0 at P=0, is seems that P is gauge pressure instead (the difference between both seems to be negligible in the range of considered pressures).

Page 4, line 136: L0 should have 0 in lower index.

Page 14, equations 11 and 12: I guess it would be instructive to write explicitly what are the units for the quantities involved, as the empirical fitting formula 11 enforces particular units.

Author Response

Dear reviewer:

Thank you for your conmments. Please see the attachment.

Reviewer 2 Report

Title: New considerations on the determination of the apparent shear viscosity of polymer melt with micro capillary dies

Authors: Wangqing Wu, Ke Zeng, Baishun Zhao, Fengnan Duan, Bingyan Jiang

  1. Each datum point should be averaged from at least three independent experiments in order to reduce the error and confirm stability of results.

  1. Data in Table 1 (kg/cm2) should be given in SI units (Pa, pascals)!

(even, units Kg/cm2 are wrongly written, it should be kg/cm2)

Also, instead of typical values, some experimentally derived data (especially MFR and density) would be also appreciated!

  1. Line 70:

However, in one accidental experi-70 ment, the precompression pressure was not set to 1 MPa, but only 0.5 MPa, and the 71 amount of polymer melt in the barrel was less than usual.”

This sentence refers to the present experiments; then, something like “data not shown here” should be added.

Or, if somewhere else presented data? Then, reference!

  1. Line 210: “shown in Fig. 2(c) and (d), …”.

Presumably, should be Fig. 3(c) and (d)

  1. Line 211: “curves at different pre-compaction pressures are also completely overlapped”.

Apparently, not completely true! Additional 3rd experiments at different pre-compaction pressures would be required.

  1. Lines 214-237:

The results are shown only for a 500μm capillary die! Results for a low sized capillaries are not shown (or the data and Figure 6 should be moved here)! Also, data should be checked for the t1/2 behavior and scaled with mass.

  1. Lines 249-257:

Possible explanations should be considered for the observed reduced filling speed than the expectations: surface roughness effects (especially as the dies are commercially produced), motionless layers due to polymer-wall adhesion interaction, and dynamic contact angle effect.

Repeated experiments could provide information on the stability of “catastrophic” increase in viscosity behaviour!

  1. Line 261: “The accuracy of the rheological measurement is..”

Here, the term “accuracy” is not appropriate. At best, delete it.

  1. Line 271-273: “However, when the amount of material is 18g, no matter what the pre-compaction pressure is, accurate rheological data cannot be obtained in the initial shear rate range (Fig. 5(e) and (f)).”

Do not use term “accurate” but, for example, stable. However, the critical shear rate decreases with the pre-compaction pressure. (Note also, that the time to reach the pressure sensor collection point is decreasing with the pre-compaction pressure, Fig. 4.).

  1. Line 283: “The polymer particles are poured into the barrel 383 above the flat plug by the conventional measurement methods.”

The polymer pellets are poured…

  1. Figure 13

It would be better to give the plot shear stress vs. shear rate as separate graph, or better describe the insert!

Author Response

(The authors gave the same response as above.)
